# Antimicrobial Tolerance in *Salmonella*: Contributions to Survival and Persistence in Processing Environments

**DOI:** 10.3390/ani14040578

**Published:** 2024-02-09

**Authors:** Tomi Obe, Aaron S. Kiess, Ramakrishna Nannapaneni

**Affiliations:** 1Department of Poultry Science, University of Arkansas, Fayetteville, AR 72701, USA; 2Prestage Department of Poultry Science, College of Agriculture and Life Sciences, North Carolina State University, Raleigh, NC 27695, USA; askiess@ncsu.edu; 3Department of Food Science, Nutrition and Health Promotion, Mississippi State University, Mississippi, MS 39762, USA; nannapaneni@fsnhp.msstate.edu

**Keywords:** antimicrobials, peracetic acid, chlorine, QACs, tolerance, antimicrobial resistance, *Salmonella*

## Abstract

**Simple Summary:**

Controlling *Salmonella* in poultry has been a top priority for the poultry industry, and while the current control measures have significantly addressed food safety concerns, *Salmonella* has continued to adapt to the environment. The question of *Salmonella* evolution and adaptation to diverse environmental conditions, especially in food processing, is important to explore in order to define effective mitigation practices. Moreover, due to the diversity in *Salmonella* serotypes, effective control strategies will have to be multifaceted, encompassing different antimicrobials with diverse modes of action and targets.

**Abstract:**

*Salmonella* remains a top bacterial pathogen implicated in several food-borne outbreaks, despite the use of antimicrobials and sanitizers during production and processing. While these chemicals have been effective, *Salmonella* has shown the ability to survive and persist in poultry processing environments. This can be credited to its microbial ability to adapt and develop/acquire tolerance and/or resistance to different antimicrobial agents including oxidizers, acids (organic and inorganic), phenols, and surfactants. Moreover, there are several factors in processing environments that can limit the efficacy of these antimicrobials, thus allowing survival and persistence. This mini-review examines the antimicrobial activity of common disinfectants/sanitizers used in poultry processing environments and the ability of *Salmonella* to respond with innate or acquired tolerance and survive exposure to persists in such environments. Instead of relying on a single antimicrobial agent, the right combination of different disinfectants needs to be developed to target multiple pathways within *Salmonella*.

## 1. Introduction

*Salmonella* remains a top bacterial pathogen of food safety concern, especially in poultry and poultry products. About 20% of food-borne salmonellosis are attributed to chicken or chicken products, and the United States Centers for Disease Control and Prevention (CDC) estimated that 1.35 million *Salmonella* infections occur each year [1,2]. The European Union One Health 2021 Zoonoses Report also confirms that the first and second most reported zoonoses in humans were, respectively, campylobacteriosis (accountable for >62% of confirmed human cases) and salmonellosis (accountable for >29% of confirmed human cases) [3]. While poultry production has continued to increase globally in the past few decades, the ability to effectively control persistent *Salmonella* populations during poultry production and processing has continued to be challenging [4,5,6,7]. Antimicrobial agents, including disinfectants and sanitizers, are commonly used directly or indirectly on products and processing surfaces to control naturally occurring bacterial pathogens and limit contamination. However, due to constant exposure and other environmental factors, *Salmonella* and other microbial populations could innately tolerate or acquire tolerance to these antimicrobials [8,9]. Antimicrobial ‘tolerance’ and ‘resistance’ are often used interchangeably, and as expressed by Maillard in 2022 [10], there is a lack of agreement among the scientific community and many studies about the term used to accurately describe bacterial reduced susceptibility to antimicrobials. Therefore, as used by Boyce in 2023 [11], ‘antimicrobial tolerance’ in this review means bacteria has shown reduced susceptibility to an antimicrobial agent.

An antimicrobial agent includes any substance or mixture that acts against microorganisms, including bacteria, fungi, and viruses. This definition could be narrowed down to a specific substance working against a class of microorganisms. For instance, an antibiotic is produced by bacteria to limit the growth or completely kill bacteria [12]. Similarly, an antiviral or antifungal is effective against viruses and fungi [13,14]. An antimicrobial agent may be used for a therapeutic purpose (to treat infection), food preservation, disinfection, and sanitization [15,16,17]. In addition, antimicrobials may be biostatic (i.e., they inhibit the growth of microorganisms) or biocidal (i.e., they kill microorganisms) [12]. As a disinfectant, an antimicrobial agent will inactivate any microorganism on an inanimate object or surface [12]. Disinfectants are commonly biostatic; they inhibit the growth of bacteria but may not necessarily be biocidal, especially on bacterial endospores [12,18]. As a sanitizer, an antimicrobial agent will inhibit the growth of microorganisms that are of the most importance to public health and can be applied on both food-contact and non-food-contact surfaces in poultry processing plants [18].

Antimicrobials used in poultry production are regulated by the USDA-FSIS and are referred to as GRAS (Generally Recognized as Safe) or safe and suitable substances. The list of the chemicals approved by the FSIS for use in poultry production can be found in the FSIS Directive 7120.1 (Safe and Suitable Ingredients Used in the Production of Meat, Poultry, and Egg Products) [17]. These are substances that may be included directly or indirectly in food products without causing any safety and quality issues. Examples of GRAS substances used in poultry processing include organic acids, essential oils, hypochlorites, peroxides, etc. A common peroxide-based antimicrobial used in the United States poultry industry is peracetic acid, and its effectiveness on various poultry-borne pathogens has been demonstrated [19,20,21,22]. Antimicrobial agents (Table 1) are commonly grouped based on their composition, molecular structure, antimicrobial activities, and mode of action.

Their mode of action highly depends on the way they disrupt some essential components of the bacterial cell such as DNA synthesis, denaturation of proteins and/or enzymes, production of ATP, and protein synthesis [12,27,28]. This review will focus on common antimicrobials used in poultry processing (Figure 1), including their antimicrobial properties and mode of action, and *Salmonella* responses to their activities upon exposure.

## 2. Peroxide-Based Compounds

Peroxyacetic acid (PAA) and hydrogen peroxide (H_2_O_2_) are the two major types of peroxide-based compounds widely used in poultry production and processing. At pre-harvest poultry production, peroxides are often used for water line sanitation and water treatment to improve water quality and control microbial pathogens [29]. During post-harvest poultry processing, PAA (C_2_H_4_O_3_) is currently the most common peroxide-based compound utilized. PAA is an equilibrium mix of hydrogen peroxide (H_2_O_2_) and acetic acid (CH_3_COOH) [30]. Currently, PAA is mostly used on poultry carcasses in chilling tanks as a replacement for chlorine. This is because the efficacy of PAA is less affected by organic matter found in cooler water, which can negatively impact chlorine [31], although as pH increases from acidic to neutral, PAA can lose its effectiveness [18]. PAA is a strong oxidizing agent with a broad-spectrum antimicrobial action on bacteria [32]. This is credited to its multifunctional activities as a biocide against bacteria and their spores, viruses, and fungi even at low concentrations [12].

PAA is very active at low concentrations in comparison to hydrogen peroxide; it produces a clear, colorless solution with a pungent odor, and it is commercially available in concentrations that range between 5% and 25% [12,30]. The USDA-FSIS permits PAA concentrations up to 2000 ppm on poultry carcasses, but it is commonly used between 100 and 200 ppm on food-contact surfaces (directive 7120.1 Rev 58) [17,28]. In addition, unlike other oxidizing agents such as chlorine, PAA does not corrode equipment, which makes it an antimicrobial of choice for poultry processing equipment [18]. Even though PAA is good for the environment due to the by-products (oxygen, acetic acid, and water) produced upon decomposition, it disintegrates easily, which makes it less stable compared to hydrogen peroxide. Another drawback to the use of PAA is cost; PAA is more expensive than other antimicrobials [33] but its strong antimicrobial activities on food-borne pathogens makes it an antimicrobial of choice in most poultry processing plants in the U.S.

PAA can function as bactericidal, sporicidal, fungicidal and virucidal [12,32]. This is due to its nature and ability to act as both an acidic and oxidizing agent [20]. Many studies have reported the antimicrobial effectiveness of PAA on both Gram-positive and Gram-negative bacteria populations [19,20,22,34,35]. King et al. (2005) [34] evaluated the germicidal activities of PAA against *E*. *coli* 0157:H7 and *Salmonella* Typhimurium on beef carcass surfaces and reported that PAA was less effective against these pathogens at concentrations up to 600 ppm when compared to 2–4% lactic acid. The authors suggested that the PAA used may have a less noticeable antimicrobial effect on the specific pathogens tested in the study compared to pathogens from other studies. Furthermore, the authors argued that the surface to which pathogens were attached as affected by temperature (chilled vs. hot carcass surface) might have a significant effect on the efficacy of PAA. Nonetheless, over the past few decades, the activities of PAA on food-borne pathogens have been observed to be highly effective [20,22,36,37]. Bauermeister et al. (2008) [20] tested different concentrations of PAA on poultry carcasses and observed that as little as 25 ppm of PAA could reduce *Salmonella* spp. This may be attributed to the acidic nature of PAA, and the fact that *Salmonella* thrives at a pH range between 6.5 and 7.5. Another study found that a 100 ppm PAA reduced *Campylobacter* by 1.0 log after 15 min of exposure on chicken skin [38]. When using PAA on chicken carcasses, it is important to ensure that the antimicrobial will not change the quality and sensory attributes of the meat [20]. Therefore, a higher concentration of 1000 ppm PAA was noted to be effective on *Salmonella* while maintaining the sensory attributes of the meat [39,40]. Further, PAA levels of 700 and 1000 ppm reduced *Campylobacter* and *Salmonella* in poultry meat and products by 1.3 and 1.5 log, respectively [40]. It is important to note that the majority of these studies tested PAA at concentrations below the maximum acceptable level (2000 ppm) set by the USDA-FSIS.

### Mode of Action of PAA

Like most antimicrobial agents, there is not much information concerning the mode of action of PAA, but some authors have credited the antimicrobial mechanism of action to be similar to other oxidizers and peroxide-based compounds [12,30]. PAA reportedly exerts antibacterial properties on the cell membrane by destroying membrane integrity while inhibiting protein synthesis [12,41,42]. The actions on Gram-negative bacteria may not be as rapid as Gram-positive bacteria because the outer membrane of Gram-negative bacteria acts as a defense mechanism but may also be the target site for antimicrobial actions [43]. PAA can attack membrane lipoproteins and the lipid layer of bacteria [23,30]. It has been suggested that PAA causes the disruption of chemical bonds, including sulfur and sulfhydryl bonds, which are found in the enzymes within the membrane [12]. This action eventually destroys the active transport system inside the cell membrane and subsequently hinders cellular activities [12]. Furthermore, owing to the oxidizing nature of PAA, it can oxidize and denature membranous proteins and lipids, leading to disorganization of the cell membrane content [23]. This eventually causes the cell wall to become more permeable to destruction [23]. Another notion about PAA is its ability to release hydroxyl radicals upon crossing the cell membrane. This would allow it to attack and inhibit pathogens by degrading their DNA or damaging membrane proteins [30].

## 3. Chlorine Compounds

Chlorine gas, sodium hypochlorite and calcium hypochlorite are some of the types of chlorine-releasing compounds [43]. Out of these three compounds, sodium hypochlorite (SH) is the most prevalent in the United States and readily available in the market [44]. In poultry processing, SH is frequently used as an equipment spray, a dip treatment of poultry carcasses, and a sanitizer for food-contact surfaces [18,33]. Chlorine is marketed as an antimicrobial agent containing 5 to 15% SH (*w*/*v* free chlorine). The USDA permits the use of SH at 50 ppm of free or available chlorine on chicken carcasses and 200 ppm of free chlorine on food-contact surfaces during sanitation [17,33]. Low cost and easy availability make SH an antimicrobial of choice for use at home and for food manufacturing purposes [18]. SH has a broad-spectrum antimicrobial action coupled with being a strong oxidizing agent [18]. It is, however, very unstable and the potency is easily influenced by various factors such as temperature, concentration, pH, and the presence of metals [18,45]. SH is very soluble in water, and it is expressed as free or available chlorine in solution [46]. Combined chlorine is another term that is commonly used to express chlorine concentrations in solution, particularly chlorine-releasing compounds such as chloramines [46,47]. The use of SH in food processing is limited due to its corrosivity to metals, of which most food processing equipment is produced, irritation to skin when people are exposed to it, and the possibility of forming a disinfectant by-product upon decomposition [18,46].

When SH is dissolved in water, it can be present in two forms: hypochlorous acid, HOCl and hypochlorite ion, OCl^-^. The presence of either of these two forms depends on the pH of the solution [46]. The pH affects the stability of chlorine in water and out of the two forms of sodium hypochlorite, hypochlorous acid is the most germicidal. Hypochlorous acid can be bactericidal and sporicidal; this antimicrobial activity is credited to the lack of electric charge [48]. It was observed to be very effective against the spores of *Bacillus* and *Clostridium* [49]. As previously stated, pH, among other factors, may determine the dissociation of hypochlorous acid into hypochlorite ions, which is less germicidal. When the pH of water is below 4, there would be more dissociated hypochlorous present in the solution than hypochlorite ions. On the other hand, at a pH above 6.5, both hypochlorous acid and hypochlorite ions are present in the solution [12,46]. SH has a greater antimicrobial activity at a high temperature and concentration with longer exposure on bacterial cell walls [50]. Conversely, the presence of organic matter can significantly reduce its efficacy [48].

### Mode of Action of Sodium Hypochlorite (SH)

The antimicrobial activities of SH can be attributed to its oxidizing effect on bacteria [12]. SH is a membrane-active antimicrobial, destroying the outer membrane activity of Gram-negative bacteria, thus penetrating the bacteria cell membrane and causes loss of permeability leading to cell death [12,48]. It is also active against cell membrane proteins similar to other oxidizing agents and disrupts DNA synthesis [51]. Other actions of hypochlorous acid include decreased ATP production at lower doses and reduction in cellular respiration at high doses, which could lead to the leakage of ions out of the cell [24].

## 4. Quaternary Ammonium Compounds (QACs)

Quaternary ammonium compounds (QACs) are composed of nitrogen bound to four organic groups [18]. QACs are positively charged surface-active agents called surfactants and are sometimes referred to as cationic detergents [46,52]. They contain hydrophilic polar and hydrophobic repellant chemical groups, which makes them a good detergent [12,53]. Some examples of QAC detergent include cetylpyridinium chloride (CPC), chlorhexidine, and benzalkonium chloride [12,52]. QACs possess broad-spectrum antimicrobial activity, they inhibit bacterial growth, and can be active against spore-forming bacteria [12,54]. They are applied at a range between 100 and 400 ppm and on an average of 200 ppm to sanitize food-contact surfaces in the poultry industry [17,28]. These compounds are very effective due to their residual effect on abiotic surfaces; this means after application; it is important to wait for the antimicrobial to dry to achieve the maximum effect on the pathogen [18]. QACs are active over a wide range of pH and temperatures [18,28]. In addition, they are very stable, non-toxic, odorless compounds that cause no stain or corrosion to equipment, which makes them effective detergent and sanitizer in poultry processing applications [52,55].

QACs are more active against Gram-positive bacteria, even at low concentrations, compared to Gram-negative bacteria [43]. This may be due to the fact that Gram-negative bacteria possess a thin peptidoglycan cell wall, an inner cell membrane and an outer membrane. The outer membrane defense mechanism of Gram-negative bacteria makes QACs and other antimicrobials less effective. Contrarily, Gram-positive bacteria have a thick peptidoglycan cell wall and a singular cell membrane that permits the effectiveness of QAC antimicrobial activities. Furthermore, the activities of QACs are significantly influenced by the length of their alkyl group chains [56]. QACs with chain lengths between 11 and 17 carbons (C11–C17) have the strongest antimicrobial properties in comparison to <C11 and >C17 [23,43,56]. It was reported that a C16 QAC was very effective in damaging the outer membrane of a Gram-negative bacterium when compared to a shorter length QAC [57]. The possible reason for this effectiveness could be due to the chain length, which was able to react with the fatty acid portion of the lipid bilayer of the bacterium [57]. In addition, alkyl group QACs with an alcohol such as alkyl resorcinol have greater antimicrobial activity due to their longer alkyl chain length, which enables it to transform into heptyl resorcinol [58]. Other authors suggest that the antimicrobial activity of QACs could be linked to concentration [43,58]. These chemical agents can act as bactericidal, bacteriostatic, and sporostatic [43,58]. At low concentrations, QACs can act as both bactericidal by completely inhibiting the growth of bacteria, especially Gram-positive bacteria and sporostatic by inhibiting the outgrowth of spores [12,43]. Although QACs are very stable and function well over a wide pH range, they are believed to be more active at a pH of 6.8 to 7.5 when in solution, especially against Gram-positive bacteria [58]. At >pH 8, some QACs are effective, whereas others have a maximum effect at a neutral or low pH [58].

### Mode of Action of QACs

QACs can bind to the phospholipid layer of the cell membrane as well as membranous proteins [11,24,59]. Cell surface permeability is greatly affected by the actions of QACs because once the compound binds to the cell membrane, it becomes easily absorbed by the cell [23,60]. The mode of action of QACs has been explored on different Gram-positive and Gram-negative bacteria [23,56]. One way to understand this mode of action is through the charges on QACs. Since QACs have a positively charged head, it is easy for the head to attract the cell membrane of bacteria, which is negatively charged. This will allow the penetration of the chains of QACs into the cells, which will cause seepage of the intracellular substance within the cell [23,61]. Other authors explained that the reaction between QACs and the surface of the cell membrane results in adsorption and toxicity, which causes leakage and damage to the cell membrane [56,58,62]. Similarly, some compounds among QACs like the alkyl groups react with bacterial outer membranes using ionic and hydrophobic interactions to connect to the lipid layer of the bacterial cell membrane. This causes a change in the membrane and eventually leads to the leakage of essential membrane contents [56]. For example, the action of two QACs (alkyl dimethyl benzyl ammonium chloride and dodecyl dimethyl ammonium chloride) on *Staphylococcus aureus* showed a significant amount of material leaked out of the cells at 9 ppm to 15 ppm and a 35 °C incubation temperature. Furthermore, the compounds caused autolysis of the cell due to the activities of RNases [56].

## 5. Antimicrobial Tolerance in *Salmonella*

Antimicrobial resistance, including tolerance to disinfectants, sanitizers, and antibiotics, is an ongoing threat to food safety. The impact of antibiotic resistance is evident across all sectors, including the poultry industry. The poultry-borne pathogens *Salmonella* and *Campylobacter* are reported to cause an annual estimate of over 600,000 antibiotic-resistant infections [63]. In fact, many strains of *Salmonella* recovered from food-borne outbreak investigations are multi-dug resistant [64]. While resistance guidelines and cutoff has been developed for many antibiotics by the Clinical & Laboratory Standards Institute (CLSI), the United State CDC, and The European Food Safety Authority (EFSA), there are no cutoff values for many disinfectants and sanitizers, including those previously covered in this review. Therefore, the ability of microorganisms to acquire reduced susceptibility to concentrations of antimicrobial compounds with the exception of antibiotics is referred to as ‘antimicrobial tolerance’ in this section. As previously mentioned, *Salmonella* has been inactivated at different concentrations of antimicrobials but due to the diversity in *Salmonella* serotypes and strain-to-strain variation, some serotypes and strains are able to tolerate these concentrations to survive and persist in the processing environment [65]. Food-borne pathogens encounter different environmental changes during food processing, including extreme pH and temperatures, and low levels of antimicrobials that could cause injury or stress [66]. These conditions can allow bacterial growth, survival, and cross contamination and although stringent hygienic practices are employed, pathogens can contaminate processing surfaces and products during processing. Moreover, while pathogens are sometimes able to persist due to an innate desire to survive, more often in food processing environments, an acquired tolerance to disinfectants and sanitizers is the reason for survival and persistence. Therefore, pathogens like *Escherichia coli*, *Campylobacter jejuni*, *Listeria monocytogenes*, *Salmonella enterica* and other pathogenic microbial populations in poultry products could adapt to diverse processing environment-related stressors, influencing their growth, survival, and persistence in the processing environment [67,68]. Many authors have reported an increase in the tolerance of these pathogens to different antimicrobials, including sodium hypochlorite and quaternary ammonium compounds due to prior exposure to sublethal levels [68,69,70,71,72,73]. Notably, antimicrobial tolerance could be influenced by genetic factors, including mutations and the presence of mobile genetic elements (i.e., plasmids, transposons, and integrons) [12,74,75]. It is critical to elucidate some of these innate or acquired factors as some of these pathogens with adaptive tolerance are able to use similar mechanisms to become resistant to antibiotics via cross-protection or cross-resistance [68,76]. In this section, *Salmonella* tolerance to two antimicrobials commonly used for sanitation in the poultry industry will be discussed.

### 5.1. Salmonella Tolerance to Sodium Hypochlorite

As previously discussed, the activity of sodium hypochlorite in water allows it to form hypochlorous acid, which is the most germicidal form of free or available chlorine. However, the nutrient-rich poultry processing environment with organic residue from poultry carcasses and water pH can influence the dissociation of hypochlorous acid, thus reducing the amount of free chlorine that reacts with bacterial populations in the environment. This has led to many studies that examined the ability of *Salmonella* and other food-borne pathogens to handle sublethal chlorine stress and the reaction such stressor(s) induce [77,78,79,80,81,82]. Many authors have reported diverse changes in tolerance in different *Salmonella* strains after sublethal chlorine exposure as expressed by higher antimicrobial minimum inhibitory concentrations (MIC’s) compared to the initial MIC before exposure [76,77,83,84,85,86]. This body of work collectively showed a range of 0.5-fold to 2-fold increases in the MIC after sublethal chlorine stress. This suggests that the exposure concentration could not destroy the bacterial cells, thus leading to cell repair. In addition, a change in cell structure including morphology has been induced in *Salmonella* and other food-borne pathogens as a response to chlorine stress [77,78,79]. While some authors have observed elongation in *E. coli* and *L. monocytogenes*, others have seen the development of rough, dry, and red colonies of different *Salmonella* strains and *Vibrio* spp. [72,79,87,88,89,90,91,92]. The rugose morphotype of *Salmonella* was expressed due to several passages through sublethal chlorine concentrations, which allowed the bacterium to become more tolerant to higher chlorine concentrations and other antimicrobials [77,79,89].

Since reactive chlorine species including hypochlorous acid are powerful oxidants that disrupt different cellular components in a bacterial cell, a remarkable response to this toxicity by the bacterium occurs through diverse stress response systems. When bacteria encounter hypochlorous acid, it causes lethal damage that quickly results in death [66]. However, exposure to sublethal doses causes damage to the bacterial cell that can be repaired, which allows for survival under extreme conditions [91]. Bacteria defenses against reactive chlorine species include the following: (1) upregulation of catalases and peroxidases, which are peroxide scavenging enzymes and are thought to protect against oxidative stress [92,93,94]; (2) upregulation of methionine sulfoxide reductase (Msr), an enzyme that catalyzes the reduction of methionine sulfoxide (MetO) that occurs as a result of methionine oxidation during the influx of reactive chlorine species into bacteria cells [94,95]. Overexpression of Msr in *E*. *coli* was observed to increase tolerance to hypochlorous acid [96], and (3) oxidative stressed cells have shown he upregulation of genes, including *isc*, *nif*, and *suf*, which are involved in the repair and biosynthesis of iron–sulfur clusters [94]. Assembly of iron–sulfur clusters is a critical step in the posttranslational maturation of iron–sulfur proteins [97]. Furthermore, there are other transcriptional factors involved in bacterial defenses against oxidative stress. These stress-related transcriptional factors are OxyR (oxidative stress regulator), RpoS (central stress regulator), SoxRS (a component of the oxidative stress regulator operon), and ArcA (Aerobic respirator control) protein [98,99,100]. Treatment with chlorine induces *oxyR* and the genes under its regulation in *Salmonella* Enteritidis and Typhimurium [94]. Similarly, genes under the SoxR regulon, including *micF*, which is a small RNA that represses outer membrane porins, were induced in *E*. *coli* treated with hypochlorous acid in phosphate buffer [101]. In addition, Wang et al. (2010) [94] found that genes that encoded chaperons (*dnaK*, *dnaJ*, *groE*, *groS*, *groL*, and *htpG*) were upregulated in chlorine-treated *Salmonella* serovars Enteritidis and Typhimurium. These genes help bacteria deal with protein unfolding and aggregation that occur as a result of environmental stress such as acidic pH and oxidative stress [102,103].

### 5.2. Salmonella Tolerance to QACs

First-generation quaternary ammonium disinfectants, including alkyl chains with 12 to 18 carbons, are commonly used in the poultry industry as a sanitizer for processing equipment and the facility. The efficacy of QACs to inactivate *Salmonella* has been previously described, but some factors including improper or prolonged storage and exposure to sublethal concentrations could negatively impact efficacy resulting in the selection of strains with reduced susceptibility. The acquired tolerance to QACs in *Salmonella* and other food-borne pathogens through sublethal exposure has been evaluated, and studies have reported different responses including changes in the MIC after sublethal exposure, cell growth, and morphology [81,82,104,105]. Mangalappalli-Illathu et al. (2008) [104] reported an adaptive response and survival of both planktonic and biofilms of *Salmonella* Enteritidis that were exposed to benzalkonium chloride. A double-fold increase in the MIC from 15 ppm to 30 ppm was observed for both the planktonic and biofilm cells following a sublethal exposure of 1 ppm for 6 days. In another study, a greater than 3.2% increase in the MIC was observed in *Salmonella* Typhimurium strains exposed to twice the concentration of the MIC after only one passage of the culture in a sublethal concentration [69]. Similarly, results from Garrido et al. (2015) [106] suggested that *Salmonella* isolates from meat samples could tolerate high levels of QACs up to 250 ppm, which may have contributed to the persistence of the isolates. Furthermore, tolerance to QACs could impact the growth rate in *Salmonella* strains as observed by Castelijn et al. (2014) [105], where QAC-adapted *Salmonella* strains had reduced growth when compared to the non-adapted strains. In addition, QACs are a type of antimicrobial that targets the cell surface of bacteria, causing disruption and leakage of cellular contents. While studies like that of Mangalappalli-Illathu et al. (2008) [104] reported alterations to the bacterial cell surface roughness as a result of sublethal exposure to QAC in *Salmonella*, others did not see any difference between the cell surface hydrophobicity of stress-adapted *Salmonella* Typhimurium and its non-adapted counterpart. This suggests that alterations to the cell surface hydrophobicity do not correlate with increased resistance to QACs [105]. It is possible that alterations in cell surface hydrophobicity do not result in tolerance/resistance to the QAC used if bacterial adaptation is intrinsic. However, the process of adaptation through gradual exposure to increasing sublethal concentrations could cause an alteration to the cell surface hydrophobicity as an adaptive response.

The tolerance of Gram-negative bacteria to QACs has been linked to the presence of genes that confer resistance to the compounds. The known QAC resistance genes are either carried on mobile genetic elements including plasmids and integrons or are encoded on the chromosome [107]. While some qac-resistance genes (*qac*A/B, *qac*C/D) encoded on mobile genetic elements are commonly found in Gram-positive bacteria, others (*qac*E, *qac*Edelta, *qac*F, *qac*G, *qac*H, *sug*E(p)) have been associated with Gram-negative bacteria, particularly bacteria within the Enterobacteriaceae family—*Salmonella* [107,108,109,110]. For instance, *qac*A/B have been found on plasmids, while *qac*E and *qac*Edelta were located on the 3′ conserved sequence of class 1 integrons [111,112,113,114]. Moreover, plasmids and integrons also carry several antibiotic resistance genes for β-lactamase, trimethoprim, and aminoglycosides, suggesting that QAC resistance in *Salmonella* can be co-expressed with antibiotic resistance [115,116]. The chromosome-encoded genes that confer resistance to QACs in Gram-negative bacteria include *sug*E(c), *emr*E, *ydg*E/*ydg*F, and *mdf*A [75,117]. While many QAC resistance genes have been detected in different Gram-negative bacteria, including *E. coli* isolated from retail meat, studies showing the detection of these genes in environmental samples are limited. The mechanism of resistance to QACs involves the efflux pump system, which contains energy-dependent transport proteins that depend on proton motive force to channel toxic material out of the cell [116,118,119]. There are different classes of efflux pump systems involved in bacterial resistance to QAC, but the major ones are the small multidrug resistance (SMR) family and major facilitator superfamily (MFS) [120,121]. Alternating the use of different biocides or different generations of QACs that have distinct modes of action can reduce the rate of development of antimicrobial tolerance to QACs in *Salmonella*.

## 6. Conclusions and Future Directions

Antimicrobial agents are commonly used to control microorganisms and promote food safety in poultry processing. These agents are used daily to disinfect poultry carcasses and parts during processing and to sanitize the processing environment (sanitation). While the initial Section 2, Section 3 and Section 4 of this review discussed the antimicrobial activities and mode of actions of the most common antimicrobials used in poultry processing, Section 5 elucidates the incidence of antimicrobial tolerance in the two most common agents (Chlorine and QACs) used for sanitation. Sanitation is one of the most important steps in poultry processing and it is vital to ensure appropriate use of antimicrobial agents during this process to prevent antimicrobial tolerance in food-borne pathogens. Since PAA is the most common antimicrobial used in the chiller, many studies have reported its efficacy on different pathogens, including those common to poultry. Thus, to stall the acquisition of tolerance in pathogens and promote the multi-hurdle approach of antimicrobial usage in processing, evaluation of PAA efficacy has been focused on poultry carcasses and parts. However, monitoring the innate/acquired tolerance to PAA is also critical, especially in *Salmonella* as one of the top bacterial pathogens implicated in poultry-related food-borne outbreaks.

This review indicates that all the antimicrobials reviewed have been effective but so is *Salmonella* evolution to evade their antimicrobial properties. The surviving populations have reportedly shown increased tolerance, requiring higher concentrations for attenuation. Continuous surveillance of antimicrobial tolerance patterns in *Salmonella* in different environments can help identify emerging resistant strains and monitor changes in antimicrobial tolerance levels. These data can guide the development of appropriate strategies, including multi-hurdle approaches to combat antimicrobial tolerance effectively. Instead of relying on the use of a single antimicrobial agent at multiple steps in processing, combinations of different disinfectants/sanitizers can be used to target multiple pathways within *Salmonella*. This approach can help overcome antimicrobial tolerance by effectively blocking different resistance mechanisms in *Salmonella*. More research is needed to understand the innate factors predisposing *Salmonella* to acquiring tolerance and examine the ways these factors can be controlled using the right combination of different antimicrobial agents. In the meantime, educational campaigns can help in slowing down the development of antimicrobial tolerance in food-borne *Salmonella* and other pathogens by promoting proper antimicrobial use, including the correct concentration and duration of application and by raising awareness about the consequences of antimicrobial resistance.

## Figures and Tables

**Figure 1 animals-14-00578-f001:**
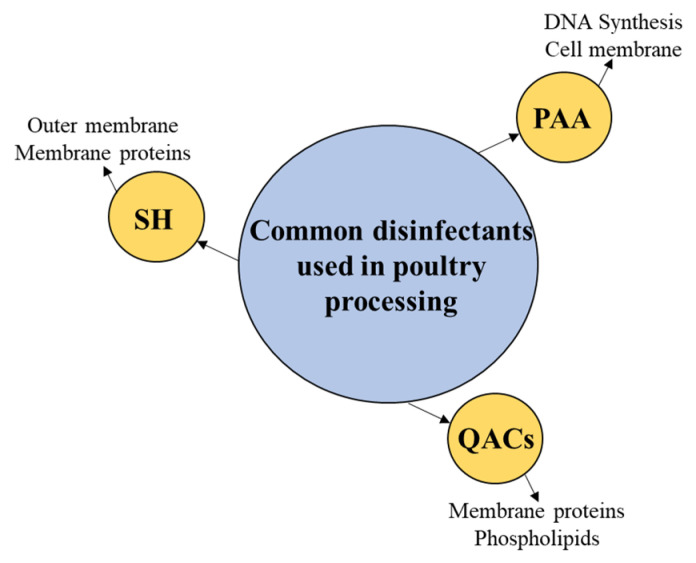
The disinfectants commonly used in poultry processing are covered in this review with a focus on their antimicrobial activity and mode of action. SH, sodium hypochlorite; PAA, peracetic acid; QACs, quaternary ammonium compounds.

**Table 1 animals-14-00578-t001:** Classes, examples, activities, and targets of antimicrobial agents [12,23,24,25,26].

Classes	Examples	Activities	Targets
Acids: Organic Inorganic	Citric acid, Acetic acid, Propionic acid, Lactic acid. Sulfuric acid, Hydrochloric acid, Phosphoric acid	Interferes with cellular uptake, affects pH gradient, and disrupts protein synthesis	Cell membrane
Chlorine Compounds	Hypochlorites, Chlorine dioxide	Protein denaturation, oxidizes peptide link and outer membrane	Cell membrane, amino group of proteins
Quaternary Ammonium Compounds (QACs)	Benzalkonium chloride, Cetylpyridinium chloride	Outer membrane damage, cellular leakage, cell lysis	Binds phospholipids, membrane proteins
Iodine Compounds	Iodine, Iodophors	Disrupts electron transport	Cytoplasmic membrane proteins
Phenols and Cresols	Ferulic acid, Garlic acid, Chlorogenic acid	Inactivation of essential enzymes, cell lysis	Cell membrane, cytoplasmic enzymes
Peroxygens	Hydrogen peroxide, Peroxyacetic acid	Denaturation, cell degradation	Cell membrane
Ozone		Nucleic acid inactivation, oxidation	Cell surface amino groups, cell
Nitrogen Compounds	Nitrite, Nitriles	Inhibition of active transport	Cell wall

## Data Availability

Not applicable.

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
