# Peer review of "Antimicrobial Tolerance in Salmonella: Contributions to Survival and Persistence in Processing Environments"

_animals, 2024, doi:10.3390/ani14040578_

Round 1

Reviewer 1 Report

Comments and Suggestions for Authors

Manuscript ID: ISSN 2076-2615

Antimicrobial Tolerance in Salmonella: Contributions to Survival and Persistence

This manuscript provides a timely review of antimicrobial tolerance in Salmonella, specifically focusing on common antimicrobials used in poultry processing environments. Salmonella remains a major foodborne pathogen, and understanding how it develops tolerance to disinfectants commonly used in the food industry is critical for improving control strategies. The topic is consistent with the scope of the journal and the manuscript is generally well structured. However, I have some suggestions I would provide to the authors:

Carefully review the manuscript for grammar, punctuation, and sentence structure issues. There are some lengthy or unclear sentences that would benefit from revision.

Expand on the public health relevance of antimicrobial tolerance in Salmonella by including more statistics/data on associated infections or outbreaks (EFSA report).

Provide more details on current knowledge gaps: what key questions need to be addressed through further research in this area?

Specific suggestions

Line 35-35: I suggest you to take into consideration the following report “The European Union One Health 2021 Zoonoses Report – EFSA” that provides comprehensive data on foodborne diseases in poultry (eg. Campylobacter, Salmonella). Please cite this paper: Pepe, T., De Dominicis, R., Esposito, G., Ventrone, I., Fratamico, P. M., & Cortesi, M. L. (2009). Detection of Campylobacter from poultry carcass skin samples at slaughter in Southern Italy. Journal of food protection, 72(8), 1718-1721.

Line 37-40: I would like to suggest that the authors consider amplifying their bibliographic research in the paper. In order to enhance the comprehensiveness and depth of the review, it is recommended to incorporate additional citations from recent and relevant studies on antimicrobial agents on different foodborne pathogens. Including a broader range of sources will not only strengthen the academic foundation of the review but also provide readers with a more extensive perspective on the subject. Please consider to add these recent works: Venuti, I., Ceruso, M., D’Angelo, C., Casillo, A., & Pepe, T. (2022). Antimicrobial activity evaluation of pure compounds obtained from Pseudoalteromonas haloplanktis against Listeria monocytogenes: Preliminary results. Italian Journal of Food Safety, 11(2); Ceruso, M., Liu, Y., Gunther IV, N. W., Pepe, T., Anastasio, A., Qi, P. X., ... & Renye Jr, J. A. (2021). Anti-listerial activity of thermophilin 110 and pediocin in fermented milk and whey. Food Control, 125, 107941.

Line 43: Please change “works” with “acts”.

Line 49: Please change "biostatic (i.e., they inhibit the growth of microbes)" to "biostatic (i.e., they inhibit the growth of microorganisms)" for consistency

Line 62-66: Could you please provide specific examples illustrating the application of these substances in the food industry? Concrete instances of their use will not only enhance the clarity of the introduction but also offer practical insights into their implementation within the food processing context (eg. the use of chlorine compounds for wastewater treatment has demonstrated their effectiveness against viruses - Suffredini, E., Pepe, T., Ventrone, I., & Croci, L. (2011). Norovirus detection in shellfish using two Real-Time RT-PCR methods. Microbiologica-Quarterly Journal of Microbiological Sciences, 34(1), 9.)

Line 79-80: Please change the sentence as follow: "Peroxyacetic acid (PAA) and hydrogen peroxide (H2O2) are the two major types of peroxide-based compounds widely used in poultry production and processing."

Line 87: "impacted" should be corrected to "affected."

Line 106: Please change the instance as follow: “PAA attacks both the bacterium and the surface to which the bacterium is attached.”

Line 131: Change along the text “peroxide based” to “peroxide-based”

Line 142-145: Please replace with this sentence: "Furthermore, owing to the oxidizing nature of PAA, it can oxidize and denature membranous proteins and lipids, leading to the disorganization of cell membrane content. This eventually causes the cell wall to become more permeable to destruction”

Line 185-186: Please replace with this sentence: “It is also active against cell membrane proteins, similar to other oxidizing agents, and disrupts DNA synthesis”

Line 191: Please change "structure of Quaternary" to "structure of quaternary"

Line 206-207: Please replace with this sentence “QACs are more active against Gram-positive bacteria, even at low concentrations, compared to Gram-negative bacteria”

Line 279: Replace "could" with "can"

I would also suggest carefully going through all the references to check for consistency and proper formatting as per journal guidelines.

Comments on the Quality of English Language

The manuscript exhibits a good command of grammar and syntax, but there are occasional grammatical errors and awkward sentence constructions.

Reviewer 2 Report

Comments and Suggestions for Authors

Dear authors,

In your manuscript you presented the tolerance of Salmonella to two different classes of disinfectant, as well as mode of actions of different disinfectant on bacteria, especially food pathogens.

The positive aspects of your manuscript are for sure the theme of this review paper- dealing with a current and important topic for many other researchers, a review of the available literature covering antimicrobial agents and disinfectants in one place; while the shortcomings are partially covered conclusions and the need for additional data in some chapters.

Please consider the change of the title – maybe to specify relatedness with poultry industry, disinfectant and similar.

Table 1 – consider to include the use of mentioned specific compounds in poultry industry and/or to relate it to specific references; consider to give at least some information why later on in the text you cover only some of the disinfectant

Figure 1 – please write the whole names of the disinfectant

Chapter 3. Chlorine compounds – here you mention calcium hypochlorite but later in the text you do not describe its role in poultry industry

L 182-183 please rephrase this sentence (oxidizing effect as a strong oxidizing agent)

Chapter 4 – please mention the use of QACs in poultry industry

Title of Chapter 4.1. please add “of”

L274 please rephrase “chlorine species”

Chapter 5. - two conclusions are not fully covered/proven in the text - the first is about the effect of disinfectants mentioned in the text on Salmonella (chlorine and others mentioned in Table 1, but not mentioned later on in details in the manuscript), and the second is "reportedly shown increased virulence" which is not supported by references in the manuscript.

Comments on the Quality of English Language

Fine

Reviewer 3 Report

Comments and Suggestions for Authors

In this manuscript, the authors reviewed the commonly used disinfectants in poultry processing facilities, with a focus on peroxide based compounds, chlorine, and QACs. Then, the authors reviewed the current evidence regarding Salmonella tolerance against chlorine and QACs. This review clearly explained the mode of action and pros/cons of the currently used sanitizers/disinfectants, and evidences of tolerance development was reviewed. However, some definitions and references were improper and should be revised. 

The major issue to be addressed for this manuscript is that the definition of tolerance is unclear. When discussing antimicrobials, the first thing that most people would think about is “resistance” rather than “tolerance” or “persistence”. Since this study focused on tolerance, it is important to make the definition clear in the beginning so that the readers would not get confused. In addition, the authors should pay attention to the references and make sure the cited references can support their statement. I found a few where the references were on the contrary to their statement (see detailed suggestions below). I did not have time to check all refs,  but I suggest the authors thoroughly re-checking all their references before re-submission. Lastly, since PAA was reviewed as a commonly used sanitizer/disinfectant, readers would be expecting to know whether Salmonella develops tolerance to that or not. Please consider reviewing related studies if present, or explain why it was not mentioned. 

Detailed suggestions are below: 

L52: Is this accurate? Disinfectants include a broad range of chemicals such as chlorine. Could you say chorine is biostatic instead of biocidal? Please check the CDC or EPA definitions regarding disinfectant or sanitizer. 

L54-56: add a reference 

Table 1: are the activities of nitrogen compounds missing? 

Figure 1: add full names of the chemicals 

L105-107: please explain briefly how does PAA attack the surface the bacterium attached. Is it the animal skin or a food contact surface? 

L113-114: Again it was not clear what “surface” means here. And if the surface can affect the efficacy of PAA in terms of pathogen disinfection, would it be accurate to say L105-107 (PAA also attacks bacteria on the surface) (if I understand it correctly)? 

L188-189: what is the estimated pH range of PAA solutions used in the poultry industry? 

L126-127: was this study done on animal carcasses and if so, was it chicken? 

L180: specify if “affect” means increase or decrease. 

L197-198: provide accurate references to support the statement of “QACs are active against spores”, or revise the sentence. Ref 55 did not study spores. And in ref 20 where you cited somewhere else, it clearly says that spore phase is unaffected by QACs. 

L202: use more appropriate references. This reference did not mention how the efficacy of QACs is affected by pH or temperature. 

L223-225: please double check the definition of sporostatic. In ref 5, it clearly says inhibit the outgrowth of spore”, which is different from what’s stated in the manuscript “affect the process of spore formation”. 

L246: what concentration? 

L263-266: I doubt if Campylobacter can grow in the environment – they are microaeorphilic bacteria and many of them can only grow at above 37dC. I agree that they can survive but probably not grow outside of the intestinal tract. 

L277: chlorin? 

L285-286: I am not sure if I understand this correctly. Does 0.5 here means 0.5 fold increase? If so, shouldn’t it mean a decrease in MIC which suggests a decreased resistance?

L360-361: not sure if this sentence is accurate. It reads like many Gram-positive bacteria do not have plasmids. And it does not read like the reason for L 356-359. Please revise. 

L382-384: increased virulence was not reviewed in this manuscript. Provide evidence in the text or consider revise the sentence. 

L386: These data. 

Round 2

Reviewer 2 Report

Comments and Suggestions for Authors

Dear authors,

thank you for taking into account all the comments given by the reviewers.

Best regards

Comments on the Quality of English Language

minor editing needed

Reviewer 3 Report

Comments and Suggestions for Authors

I appreciate the authors effort improving this manuscript. I have no further questions.